# Visibility Model of Tangible Heritage. Visualization of the Urban Heritage Environment with Spatial Analysis Methods

Elif Sarihan

Department of Civil Engineering, University of Debrecen, 4028 Debrecen, Hungary;
elifsarihan@mailbox.unideb.hu

**Abstract:** The methodological approach of the study proposes an innovative yet adaptive way to define and preserve heritage sites and their elements. In the case study, the proposed methodology guides the design/planning research of heritage sites by linking the perceptual behaviour with the information of the built environment. Visibility is the tool to measure the level of exposure of specific urban elements from a particular perspective. While isovist analyses define visibility in the built environment, fields of view from the periphery of heritage sites are applied to calculate visible or invisible areas by the observer. The purpose of the current study is the evaluation of the identification of the elements to be protected, by modelling both the heritage environment and the heritage elements according to the visibility criteria. For this purpose, I illustrate my approach by using visibility analyses and Space syntax analysis in the case of the Sulukule neighbourhood, the leading renewal project, in Istanbul. This area used to have notably cultural–historical assets–historic land walls, the lifestyle of Roma people—but now the renovation works carried out in the Sulukule case study site have affected the identity of the "visible" and "known" space of the historic quarter.

**Keywords:** visibility analysis; isovist; field of view; urban heritage; built environment; Istanbul

## 1. Introduction

The concept of heritage derives from the fact that humans create urban environments and form urban patterns as they bring together components, constantly changing their surroundings [1]. Throughout history, people have marked the places where they live with distinctive features that contain/carry information. These are the tangible patterns, such as architectural and urban heritage, and physical and historical remains as architectural and historic values [2]; or intangible patterns, such as language, belief, and traditional forms of expression of these places and given to the objects that shape contemporary historic urban landscapes [3]. Therefore, historical cities have become important places where the collective values of tangible and intangible heritage can be found and represented. Heritage sites continue to exist in the complexity of contemporary cities as remembrances of the past [4]; as part of the present urban fabric, they will persist in the future.

Spatial–temporal and natural, cultural, and social processes construct the historical urban landscape. As a result, the concept of the historical urban landscape provides a mindset to understand the urban context. The concept is related to the built environment and cultural values, as they cover local knowledge such as physical layers, intangible cultural heritage and value perception, building practices, conservation, and management, which have symbolic significance [5].

According to the definition of historic urban landscapes, the protection of cultural heritage sites plays an essential role in preserving the built environment of these areas [6]. Thus, the perceptions of urban complexity are brought together holistically, combining the tangible and intangible patterns of heritage with the layers of the built environment. Increasing unprecedented urbanization and structural transformations have profoundly created more pressure over the past decades, not only on suburban outskirts and the

inner core of the cities [7], but on societies, also affecting natural, historical, cultural, and archaeological heritage sites.

The concept of urban heritage has a global reach, with numerous definitions and contexts to which it relates. According to Olsson [8], urban heritage does not only subsume designated protected areas or heritage monuments and areas. However, it is a system set in which these values are defined in the broader environment and describe the interaction between the parts of a system where the urban landscape is described as heritage. Therefore, urban heritage should be evaluated within the system of both intangible and tangible characters. In this context, urban heritage should include tangible characters (physical remains) and less tangible characters (human beings as significant factors for the articulation of the heritage space and the built environment) [2]. Blake [9] defined it in the context of these two features, as cultural heritage brings together elements of seemingly ordinary characters, such as features of the natural and cultural landscape.

Even preservation and development of cultural heritage are target areas of international urban planning policies, but there is an emerging need for planning studies of heritage protection areas that will predetermine the results of planning stages and predetermine the solutions that will preserve heritage. These are becoming more and more important in the lines of heritage conservation and development of heritage sites. In contrast, various obstacles [9,10] prevent the visible scene of such heritage sites in the historic urban landscape. These obstacles could be the increases of newly added buildings, and the deficiencies of the spatial configurations and designs.

The current research methods offer directions to define the heritage surroundings quantitatively in historical landscape areas preserved in urban settings. Based on traditional urban morphological approaches (Conzenian [11], Caniggian [12], Gordon Cullen-Townscape [13]), the researchers created comparative urban analyses using the areas' landscape units or character zones, in combination with photos and sketches, to determine the historicity of the urban landscape. Despite that, human perception is the fundamental link between human and built environment [14]. These tangible patterns of the environment are defined by perceptual features and manipulate themselves.

The listed buildings or heritage environments are attractions of human interest because their views, meanings, and patterns reflect their cultural and historic natures [15]. However, this phenomenon is related to the visibility/perceptual conditions of these elements in the environment. The idea, generated from the research of Michael Batty [16], is related to our perception, based on the geometrical properties of different urban spaces. In terms of visual representation, it is about creating a visual field derived from any viewpoint of an observer and based on the extensive geometric properties of the environment. Batty offered an isovist analysis method to present space with spatial and statistical value. The isovist (fields of view) obtained by making use of the properties of the form or (urban and geographical) morphology [16]. Michael L. Benedikt defined isovist as "The set of all points visible from a given vantage point in space and with respect to an environment" [17] (p. 47). Based on this definition, it is possible to generate a defined visual field of the spaces and features from an observer's point of view in different points in space. The isovist and isovist fields reveal the meaning of clarity, preservation, specialist, dynamics, or complexity by human behaviour and human cognition [17]. David O'Sullivan and Alasdair Turner [18] generated the reciprocal visibility of a series of isovists from different positions and aimed to derive a general visibility graph in the space. The holistic approach, a graph-based analysis produced locally and globally, presents a methodology for defining the configuration based on accessibility and visibility [18].

Hillier and Hanson [19] used spatial analysis methods to examine how spatial structure promotes human behaviour, coexistence, and contact with humans in culturally diverse built environments and various historic environments. Analytical tools that seek to explore the visual qualities of three-dimensional space have begun to emerge in recent years [20]. Current studies on the urban environment focus on problems related to the visibility of landmarks and heritage sites within urban systems. Phil Bartie [21] studied the visibility of

landmarks within heritage systems throughout a series of visual metrics. André Soares Lopes et al. [22] studied the elements of the urban landscape that appear together as an overview of visibility analysis. They analysed the other elements that are co-visible in the visible scene by focusing on the visibility of heritage elements.

The research problem is how to visualize the outputs of the elements related to the perception and clarity of heritage sites within complex systems, and how the spatial configurations of the buildings of the heritage sites must be controlled and designed so as to define the performance of the urban elements holistically. It is significant to decide on the components integrated with heritage, or which factors are not well determined in terms of preservation of the heritage settings.

The current methods of spatial analyses lead towards understanding the urban landscape as a whole by using urban geometry as data, defining and reflecting all elements visually. The methodologies provide the adaptation of deep learning of the two-dimensional spaces to evaluate the elements that must to be protected or included in the heritage sites.

This paper offers a spatial analysis method to overcome spatial configuration problems of heritage sites with urban systems. Most importantly, the method provides a way to connect the results of integrated visible models of the urban landscape—created by multiple isovists—by learning from the characteristics of the heritage environment. Thus, while defining heritage pertinence targets within the built environment, urban designs and projects can be implemented in heritage areas with a more holistic approach.

## 2. Background

According to Athos Agapiou et al., there is a need to track innovative ways and analyse new and practical approaches to urban heritage sites away from the archaeological approach [23]. For this purpose, this research represents a method for establishing interoperability and a methodology to reduce the obstacle elements of heritage visibility. The combined approach is based on integrated visibility analysis to determine traceable results of the compositions of urban heritage settings.

Visibility is an analysis tool that provides a significant advantage in visualizing the integration of the urban environment based on visible data of fields and objects through a point of view and isovists at a particular location [24]. Developments in computer science have allowed visibility analysis to become a widely used research method today [25], including GIS-based view analyses, ArcGIS, and 3D Analyst; or spatial analysis techniques such as the space syntax based on Social Logic of Space [19]; or isovist analysis [17] used for evaluating urban or architectural spatial scenarios.

The current research aims to demonstrate the spatial compositions of the heritage environment and the different visual elements of its configurations. It includes space syntax analysis to reveal a range of people's visual directions about the landscape [26] and the complexity of various factors.

The integration of the visual analysis methods creates a more holistic approach to the representation of heritage sites. space syntax focuses graphically on the spatial arrangement of buildings and cities and how human movements affect their social and environmental consequences [27], whilst isovist analysis tends to focus on people's social experiences and perceptions in space and determine the scope of vision in the built environment [28].

The difficulty in perceiving urban space depends on the variation of shaping elements of the built environment and the isovist behaviour (an observer makes the decision). Urban spatial research must include both analyses of visibility and permeable visibility. Thus, in this way, it can be decided which spatial configurations inhibit urban elements to consider or ignore [29], and the changes of visual permeability will reveal the heritage elements.

In the current research, the focal point is on the deep/comprehensive visibility analysis. Both demonstrate the visibility character of heritage sites, enable their integration to other urban spaces, and reveal the geometry/layout of the urban heritage environment. Deep spatial learning is limited to the neighbourhood level in which the heritage environment is located. Isovist analysis is successfully used in two-dimensional spatial computing at

the neighbourhood scale. It is beneficial in analysing the degree of visibility of landmarks (heritage sites, listed buildings, monuments) or the panoptic appearance of these areas as they move through space and determine how urban interventions will affect these elements [30]. The visibility of the observed area and the amount of appearance vary concerning the diversity of the area around the space [31]. According to Y. Kim and S. K. Jung [32], the isometric measurement approximates the amount of visual information at a given point, and the isovist field reflects the amount of visual designated field from observation points. Visible areas inside or the Field of View (FOV) may not be visible outside of the space due to the permeability of the geometry. In the historical landscape of heritage sites as the effective environment with visible elements, we should consider all the elements that can be seen simultaneously with historical elements from all possible points of view. The space establishing elements are distributed in different positions and combinations based on the observer in a given point of view and a given horizontal Line of Sight [33] (Figure 1). Therefore, we must decide which elements can be sequenced in the formation of visible areas. While it is difficult to isolate what will be preserved (buildings, urban ensembles) in such complex systems [34], it is clear that the unique identity of the heritage environments must be maintained.

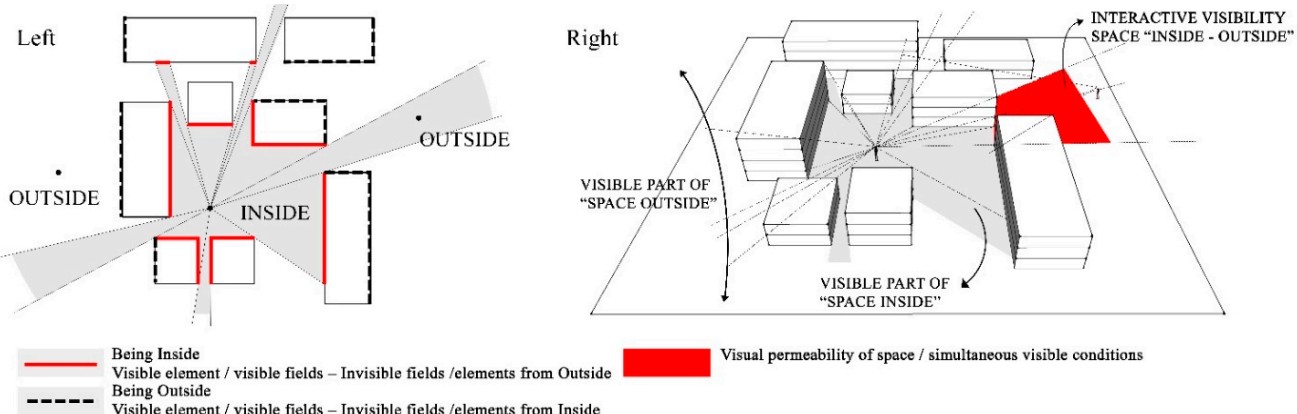

**Figure 1.** The different visibility conditions based on the position of the observer, inspired by Philip Thiel [33] (pp. 222–224). Left is a space bounded with physical boundaries; Right is a visual environment in Left; re-illustration by the Author.

The Field of View (FOV) in landscape architecture is a concept dating from about the 1960s and has been adopted by many disciplines. It displays the areas visible from a single point of view by measuring the Lines of Sight (LOSs) from that point to all other localities in the working area [35]. According to Phil Bartie, the Field of View (FOV) is measured as the most large-scale observable horizontal angle among targets in the most certain Field of Interest (FOI) created from a given viewpoint. It derived from object size and orientation measures but does not determine the angle vision that may be obstructed from the inside of the FOI [21]. The front area is often measured by placing the apparent scope under each target, taking the unknown obstacle areas between the observer and the target into account (Figure 2).

The main purpose of the current research is to evaluate the various methods of visibility analyses combining the constituent elements of heritage sites, and possible/already established design and planning indicators in their integration with the built environment to provide a framework based on visibility analysis on the visual preservation of heritage.

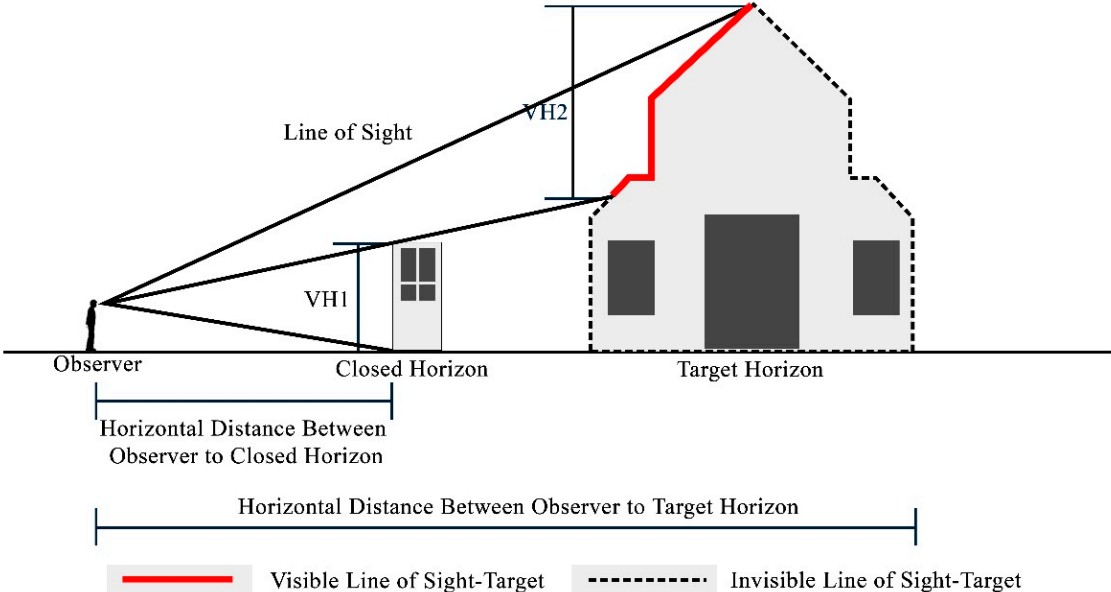

**Figure 2.** Visual representation of LOS. Closed horizon location indicates the level of visibility on the target horizon; inspired by Bartie et al. [35]; re-illustrated by the Author.

## 3. Materials and Methods

For seeing perceptual patterns in heritage sites through the generation from any point where an observer establishes a different visible field (isovist), the research combined different approaches.

Firstly, to generate the visible field, we had to determine the phenomena, namely an outline of the spaces/nature of the isovist boundaries and points of view. The isovist analysis was applied according to Benedikt [17]. The shape and size of the isovist are determined based on the environment's geometry; for this purpose, Batty's research [16] was considered, and the research used the methods of Bartie et al. [21] and Lopes et al. [22] to determine the object or size that obstructed the visibility conditions of the field of interest (heritage target).

Lastly, to evaluate the mutual visibility of heritage elements and the built elements of their surrounding built elements, visibility graph analysis was conducted for each selected heritage site, according to Turner et al.'s research [29].

### 3.1. Methodology Overview

The methodological approach proposed a new and general way of defining the heritage site and preserving heritage elements. In the case study, the methodology guided the design/planning study of heritage sites by linking the perceptual behaviour in the heritage space with the information and attention of the urban heritage environment. The types of information about the visual experiences of the space, generated with the visibility analysis, are based on the observer's point of view. Physical information/tangible data (such as location, boundaries, or dimensions of the heritage site) are real life data and do not allow any judgment about their existence or nature. Intangible data are flexible and allow for some value judgments and are related to the are non-quantitative cognitive and perceptual aspects of space and properties of objects.

Therefore, a distinction of interest had to made between these two elements in the current research. Each heritage site shows diversity and different physical patterns that have different information [36].

The framework stage used for the case studies is presented in Figure 3 and includes an outline of the data, separating the data of the sites, and moving towards the more detailed resolution, step by step. It starts with the identification of real data followed by

the selection of a set of isovists that generates the spatial system with all its elements. The final step is to evaluate all visibility conditions with graph analysis (Figure 3).

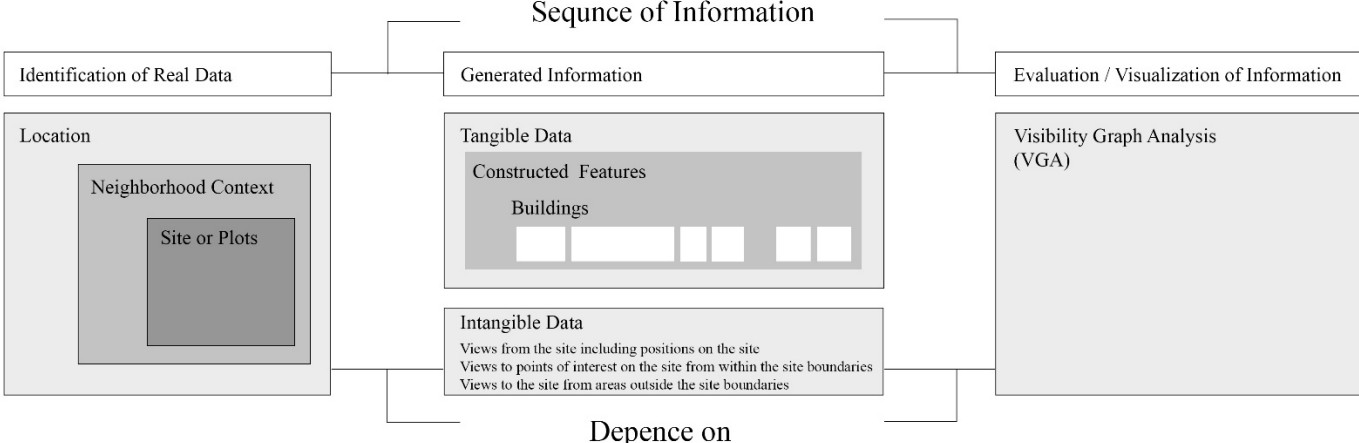

**Figure 3.** The framework of the proposed stages.

*3.2. Identification of Real Data*

The first stage of the identification part of the model (A) involves mapping the heritage site's location with respect to the city as a whole. This type of mapping method allows planners or decision-makers to understand the heritage space or to create information-containing elements. The neighbourhood context (B) represents the immediate surroundings of heritage sites beyond one or two blocks away. Thus, all conditions that may affect the heritage site can be evaluated and displayed together. In heritage sites, we need to determine the "outline of the spaces"/"nature of the isovist boundaries" (C) in order to create isovists (visible area) in the generation of perceptual information from any point where an observer will create a different visible area (isovist) (Figure 4).

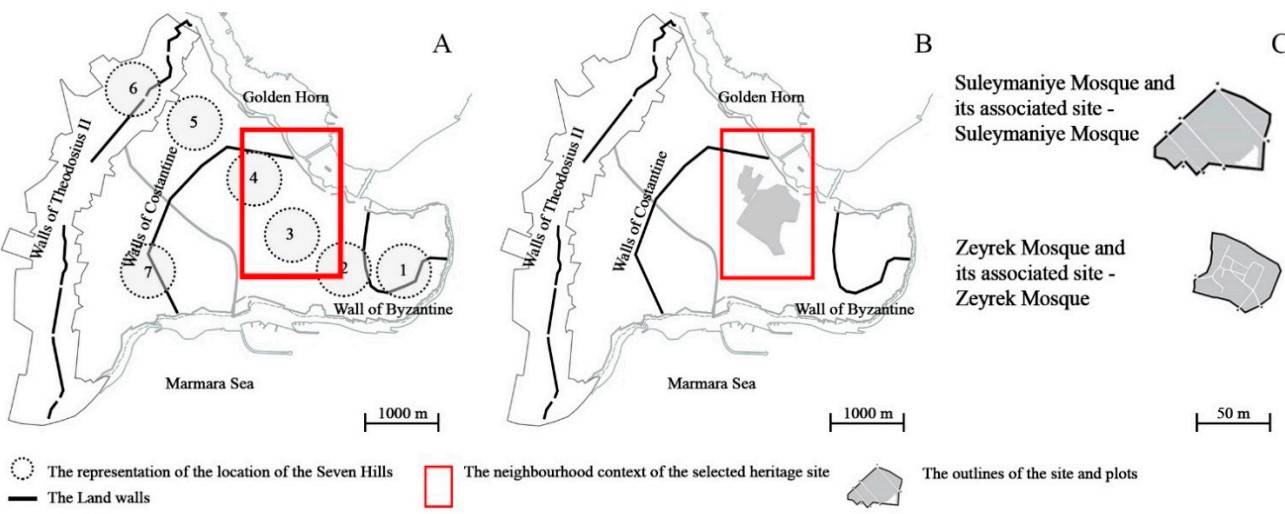

**Figure 4.** Mapping real time data for identification of heritage sites and immediate surroundings is based on OpenStreetMap and created by the Author.

Tangible patterns add more meaning to the appearance of heritage, thus making it possible to make more detailed inferences to perceptual attributes on the spatial level.

*3.3. Generated Information*

3.3.1. Constructed Features

Tangible elements of heritage sites are created by revealing blocks and primary building volumes. Assets of all urban components are determined by evaluating the heritage sites' tangible data. Due to the lack of information visually presented by the two-dimensional analysis, three-dimensional forms create [37] the existing perceptions of the spaces at this methodological stage. They also contribute to the analysis of the dominant current architectural character surrounding the heritage sites. According to Lopes et al [22] the mapping method of tangible patterns shown in the methodology allows planners or decision-makers to understand the three-dimensional approach (e.g., 3D landscapes, visibility analysis, and eye ray tracking analysis) or focus on the two-dimensional analyses (complementary analysis and visibility measures) of the spatial attributes of heritage sites and complementary interpretations about visual features (Figure 5).

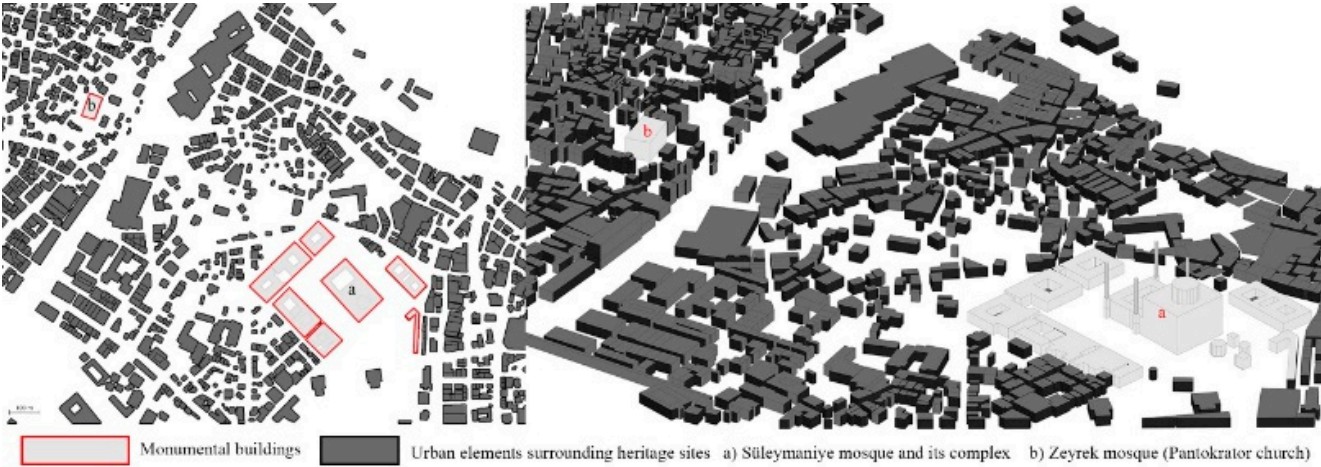

**Figure 5.** The constructed features of heritage sites and existing architectural monumental buildings that characterize their surroundings; based on OpenStreetMap; illustrated by the Author.

3.3.2. Intangible Data

This stage identifies the points in the heritage sites where the observer's field of view is closed/blocked or open. These obstacles are buildings, blocks, or walls according to the tangible pattern of the urban environment defined above. Visible fields or visibility conditions of the heritage surroundings cognitively describe the space. Therefore, it is possible to predict human perception and behaviour (intangible). Isovists take the shape of the environment or geometry of space. Thus, our way of experiencing a field and our visual perceptions are related to the isovists.

Since the purpose of this stage is to identify intangible patterns of the environment of the heritage sites, it should reveal the meaningful features of the environment related to human perception, although intangible heritage should be defined as spaces that are used and perceived by individuals.

In this stage, we generate the point of view from outside the boundaries of these elements, even including components of the urban surroundings of the heritage site. Thus, all possible combinations can be determined regarding visibility conditions within the visible cluster, visible angles, visible permeabilities, the most dramatic/less visible positions, or highly visible areas of the heritage surroundings.

The solution is the chosen perspectives or points where the targeted heritage buildings are visible/or not visible.

The number of visible rays in each sight frame varies according to the perceived distance, size, and form of the obstacles. For this reason, viewpoints were obtained from

the periphery of the heritage site. The effects of the urban elements on the visibility of the target and the visibility behaviour of the distance were determined (Figure 6).

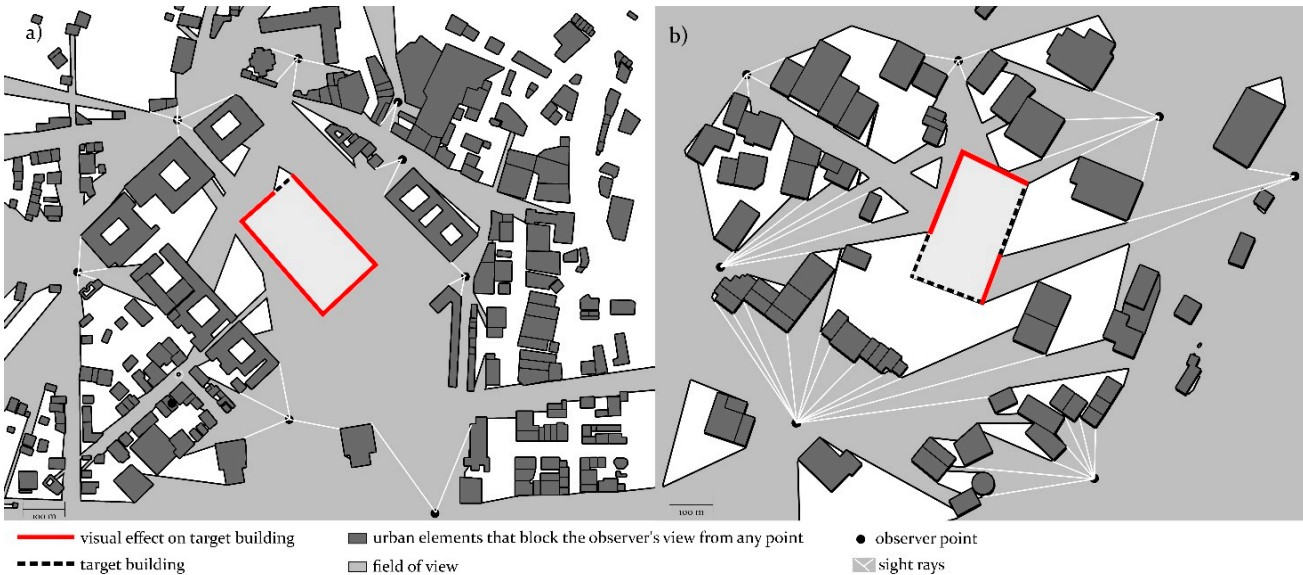

**Figure 6.** The isovist rays respond to the visibility conditions of the surrounding heritage and visual permeability from inside and outside of the space. The perimeter of the historical building targeted/point of interest as the Süleymaniye Mosque in (**a**) was determined by 2D analysis. The historical building environment targeted/point of interest as the Zeyrek Mosque in (**b**) was defined by 2D analysis, generated on the isovist platform [28] by the Author.

### 3.4. Evaluation/Visualization of Information

Visibility analysis tools determine the perceptual qualities of architecture or the built environment and characterize different urban system types as a whole [16] based on matching multiple visible criteria.

Heritage areas, especially cultural heritage sites, can be evaluated by the spatial-based framework, by combining it with its cultural and social framework [18]. In the consideration and evaluation of heritage, it is important to combine the interaction between traditional land use, relevant social characteristics, mobility, and interests.

## 4. Case Study

Istanbul's Historical Peninsula served as the capital city of several civilizations, such as the Roman, Byzantine, and Ottoman Empires. It hosts the oldest settlement in Istanbul. It was the capital of the Byzantine Empire for 1058 years, and then the Ottoman Empire conquered the city and hosted the Ottoman Empire as its capital for 469 years. Istanbul is situated on the northern part of the Marmara Sea between the Bosphorus and the Golden Horn natural harbour. Its topography consists of hills overlooking the water (Seven Hill Istanbul), slopes, and valleys heading to the shores and valleys. Located on the first hill overlooking the Golden Horn, the Acropolis was fortified with the Sur-i Sultani, after the Ottoman Empire came under sovereignty, and Topkapı Palace (New Palace), allocated to the state administration under this inner castle, was established and has been the administrative centre of the Ottoman Empire for centuries. An understanding of urban zoning in the Ottoman Period, considering the topography, aims to position the wide range of service structures (complexes) on the hills to reflect in the view the hierarchy between these buildings, and at the same time to point out the centres of the sub-regions/districts of the city and to provide an understanding and orientation of the city in today's words. The outstanding universal value of Istanbul reflects the unique incorporation of culture and characteristics of several civilizations experienced in the city, overlapped on the city, and shaped with its unique silhouette the social and physical patterns visible today [38].

In the Historical Peninsula, the historical strata are multi-layered within the contemporary urban structure [39]. The monumental structures, existing in the same urban areas of the Historical Peninsula conservation site, reflect a transitional character in the urban fabric with the changing socio-cultural features. The backbone of the urban form was shaped according to the city's topography in specific periods. Important religious buildings or monuments developed and transformed [40], such as churches from the Byzantine period and mosques from the Ottoman period [41], and their symbolic meanings sustained. Over time, the Historical Peninsula partially lost its character. As an example of changes, the historic walls, which were the most important symbol of the Historical Peninsula, whose construction was started by Theodosius in the beginning of the fifth century, and the Top-kapı region (included in the UNESCO World Heritage list) provided the defence function but also determined the size and development of the city. However, while some of the walls have survived until today, some have been demolished.

The Historical Peninsula has four areas that carry outstanding universal value, as one of the criteria to be listed on the UNESCO's World Heritage List (listed from 1985): the Archaeological Park at the tip of the Historical Peninsula; the Süleymaniye Quarter with the Süleymaniye Mosque complex, bazaars and vernacular settlement around it; the Zeyrek settlement area around the Zeyrek Mosque (the former church of the Pantocrator); and the area along both sides of the Theodosian land walls including remains of the former Blachernae Palace [42]. Monuments are known as unique architectural masterpieces of the Byzantine and Ottoman periods, such as the Süleymaniye Mosque designed by Mimar Sinan, the Hagia Sophia church, the presence of historical Byzantine walls, the mosaics of the palaces and churches of Constantinople influencing both Eastern and Western art, and the local residences around the important religious monuments in the neighbourhoods of Süleymaniye and Zeyrek. Elements such these that reflect the remains of the Ottoman urban texture have caused these four areas to be included in UNESCO's World Heritage list in 1985 [43]. The topography of the Historical Peninsula offers views of the city from many angles, including its seven hills; from the inner parts of the peninsula, it is possible to capture the scenery and even the sea. Those seven hills listed make it possible to see potential views that have overshadowed the city's skyline (Figure 7).

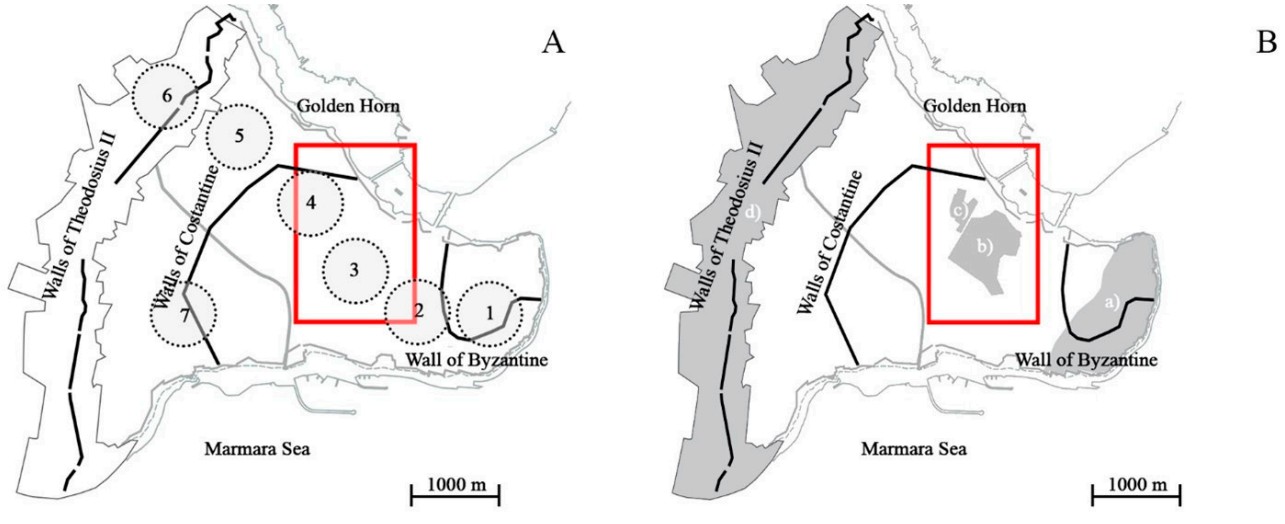

**Figure 7.** A representation of the seven hills of Istanbul (**A**) and the Historical Peninsula World Heritage sites (**B**): (b) Süleymaniye and (c) Zeyrek case studies map; map based on UNESCO [43,44]; re-illustration by the Author.

The Süleymaniye World Heritage case study site is situated on the third hill and continues towards the shores of the Golden Horn. The district shows the typical characteristics of the Ottoman Era settlement with its traditional houses and neighbourhoods formed by the streets, preserving their organic forms. The main element of the district is the Süleymaniye Mosque and the secular urban fabric around it.

The second case study is the Zeyrek Mosque World Heritage Site, located on the fourth hill and hillside of Istanbul, bordered by Atatürk Boulevard to the east. Atatürk Boulevard separates it from the Süleymaniye District (and the Süleymaniye Mosque and its Social Complex). The Zeyrek District is known as the fourth hill of Istanbul and was recognized as the monastery zone during the early Byzantine period [42]. Its traditional fabric is preserved, which consists of timber attached buildings that reflect the residential area's characteristics.

Zeyrek and Süleymaniye Mosques belong to different religions, cultures, and communities, located on two hills facing each other in the Historical Peninsula. From the Byzantine period to the Ottoman period, the two masterpieces positioned on these hills created a remarkable visual impact from many points in the cultural heritage site. Although these areas were designated as conservation areas from 1995, no conservation-oriented development plan was prepared for the Historical Peninsula until 2003. Since 2003, incomplete and inconsistent planning processes [45] caused the limited implementation of the conservation-oriented development plan. A comprehensive legislative structure (Law No. 5366) ("Law on conservation by renovation and use by revitalization of the deteriorated historical and cultural immovable property") [46] has been developed to undertake urban renewal initiatives in Istanbul's historic neighbourhoods. Istanbul's planning experts and scholars criticized the legislation for the social aspect of urban development, exclusion of residents, infringement of property rights, and negligence. Despite these objections, the Law entered into force. The legislation provided the basis for new urban renovation initiatives in the historic quarters of Istanbul. New legislation granted further responsibilities to municipalities for undertaking renovation programs in historic neighbourhoods. Therefore, World Heritage sites have been affected by urbanization processes due to their acceptability for investments in the tourism and housing sectors. Due to the limited intervention of UNESCO in these areas at the local level, the protection of historic neighbourhoods such as Zeyrek and Süleymaniye remains insufficient [47]. Furthermore, the poor design of the existing neighbourhoods and the construction of the densely built environment prevent a strong interaction of these Byzantine and Ottoman icons. Although they are close to each other, they are perceived far away and are disconnected by the human eye-level experience.

## 5. Implementation of the Method on the Renewal Areas

The purpose of the renovation zones declared per Law No. 5366 is to plan the neighbourhood where the historical–cultural heritage properties are concentrated. This policy has predominantly impacted Istanbul and its 47 historic districts. The leading examples of the renovation areas within the historical borders of the peninsula are the Süleymaniye, Sulukule, and Ayvansaray districts. These areas are occupied by middle-and low-income groups in the city centre. However, the planning decision aimed at improving the urban standards of selected regions with the physical renewal and quality of life of the settlers could not interfere with the accompanying social and economic processes. As a result, the former inhabitants had to migrate from their original settlements (gentrification) because they failed to adopt new economic and social conditions [48]. The investments focused on these renewal areas for economic development because these neighbourhoods had great opportunities to transform to satisfy the requirements of tourism, offices, and residences in heritage surroundings. Cultural heritage values understood as tangible and intangible features of heritage sites are significant elements of cities. Therefore, the protection of these elements should be taken into account when planning interventions in these areas. Now, with the renewal projects being carried out in leading historic communities like the Sulukule neighbourhood, attention is drawn to the importance given to those areas. The

Historical Peninsula has a multi-layered structure with historical changes and development. Among them, the Sulukule neighbourhood lies along the land walls of Istanbul, elements of the image of the city [49].

If the cultural heritage concepts are not defined, the heritage of regions may face pressure because of new developments [50]. However, multi-layered heritage neighbourhoods and structures can be maintained and controlled through planning and design decisions. The renovation works carried out in the Sulukule case study site affected the identity of the "visible" and "known" spaces of the historic quarter. In this context, this study follows the examination of how the renewal process affects cultural–historical assets by using visibility analyses and space syntax analysis in the Sulukule example of a renewal area.

## 6. Results

### 6.1. Visibility Graph Analysis (VGA) Model—The Spatial Connection with Urban Environments

According to Turner [29], researchers, planners, or architects can analyse a visibility map for a spatial context by using some of the many metrics developed to examine graphic features across a spectrum of disciplines, thereby providing insight into the range of available measures.

The visibility graphs of the two case studies in the heritage sites were developed using isovist software. These are semi-local or relational measures that extend between local and global information: visibility, mean metric depth, mean visual depth, and integration. A planner's isovist field of visibility and accessibility created from a specific point to establish a network of all direct connections between nodes. The values of the visibility measurements in the analysis were represented using the colour scale in each analysis type. Furthermore, the extent of the analysis as well as the visibility of case studies in the heritage area, plotted using a scale from blue (minimum area) to red (maximum area) for a simple spatial configuration, was represented.

The red–blue colour spectrum visibility graph reflects how frequently a field falls into an isovist generated from within that area. Figure 8 shows a graphical comparison of the study areas based on the results of the space syntax analysis. According to the results of VGA, the first proposal indicates the most visually integrated, and the shallowest nodes on average are shown in red, while the least visually combined and deepest nodes on average are depicted in blue. The core of the two study areas and the part of the Atatürk Boulevard close to the cores are not visually integrated. The lack of circulation of pedestrian areas that would connect the Zeyrek Mosque and the Süleymaniye Mosque areas affect the integration in the region. It is clear how the importance of the correlation between visibility analysis and connectivity and integration provides clues to users in the entire spatial configuration [51].

The second proposal of visual metric depth analysis shows that the observer's location (point) in the historical site is the shortest metric distance from that point to a single global position. The red colour means that the metric depth is the longest path distance from a specific location of the observer to a global sample location. The third proposal of mean visual depth in the plane is presented for the illustrated number of visual measures from the point to all locations. The visual step depth determines the pedestrian flows between different routes in the spatial configuration and the quality of pedestrian accessibility and the accessibility to public spaces or public services [28].

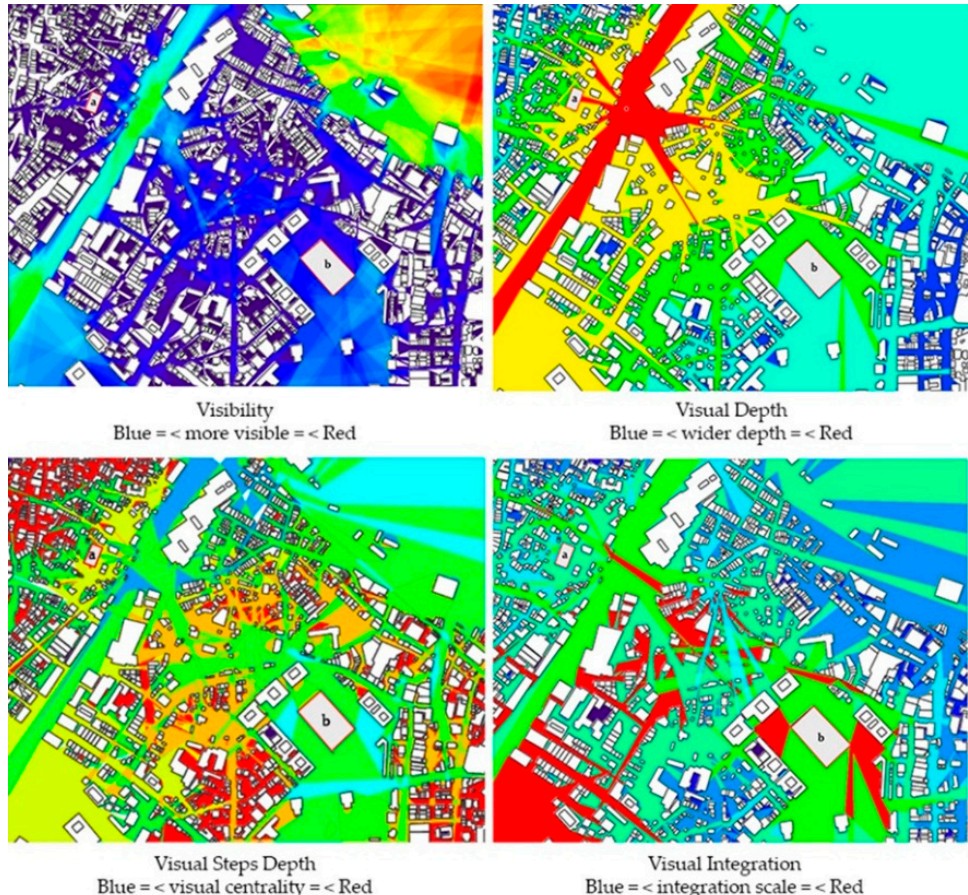

**Figure 8.** Visibility analysis results of the selected heritage sites, generated on the isovist platform [28] by Author.

In this context, the results obtained show that the visual step depth metrics were not sufficient for the accessibility of cultural heritage sites. The integration, which is proposed as last, is about the average number of lines required to go to all areas in the spatial system, not accessibility as a metric, but depth [52], and is used to show how far a particular area is from another area. In addition, integration is typically indicative of the number of people likely to be in a space [53]. In this context, red represents the most integrated spaces in the heritage site, while blue represents the least visually integrated areas from all other nodes. The result in the visual integration analysis is the integration surrounding in the core of two cultural heritage sites. It consists of the combination of primary and secondary integration areas and the central integration of the Atatürk Boulevard.

In this analysis, it was determined that the cultural heritage sites in urban areas and the spatial configuration of the heritage elements within them, land use compositions, and characteristics of the urban form affect the visibility and accessibility of the heritage site in many ways. Moreover, the variability of land uses around the cultural heritage site and the increased spatial intertwining between building densities indicate that space has a strong effect on regional accessibility behaviour.

### 6.2. Mapping the Visual Configurations of Spaces

These measurements, created by combining the GIS datasets of the Zeyrek and Süleymaniye study areas, can provide a user with the ability to be context-sensitive to the historical heritage area. When searching for spatial databases, visibility criteria can rank the results that show the most visible objects [35]. In navigating with visibility maps, visible field values (greyscale values) can guide the user to good viewpoints in the field. When an observer travelling from the settled field experiences visible field changes (corner to centre),

the visible metric values change (light grey means a less visual distance to all other points), and the total duration of the trip increases (light grey means less metric distance). We can determine the measures of the visibility of the cultural heritage structures, regardless of the distance, and how many of them emerge or do not emerge as they move away from the structure.

Results can be derived from maps, for example, to perceive the target of the Zeyrek Mosque, and functions such as (1) which direction the observer should move, (2) visibility depths, (3) visible functions (visual connections, visual dominates), and (4) spatial integration can be calculated. However, as the observation distance increases, the perceived area decreases. This means that there is more exposure to a large (Süleymaniye) area and a small (Zeyrek) area than two equivalent areas. Even if the total area is equal, a large area and a small area may appear larger than two equal-sized areas [17]. According to Bill Hillier [20], human perception of space and time is positive in this sense. The Süleymaniye Mosque and the Zeyrek Mosque and the historic neighbourhood pattern that developed around them give heritage value to the cultural heritage site. However, in recent years, the World Heritage values in the Historical Peninsula have been negatively affected by the intense and linear housing pressure caused by rapid urbanization.

The perceived interest and visual impact of the Süleymaniye Mosque and the Zeyrek Mosque from many existing points has decreased. Figures 9 and 10 presents the findings of the visual perception analyses of the Süleymaniye and Zeyrek heritage sites. Results show that the fields of view around the mosque complexes have become relatively smaller. This is linked to the low amount of visually perceived dominance. In the analysis, visible metrics around the heritage buildings indicate that the visitor travelling in the cultural heritage sites experiences changes in visible areas, and the visible values of the heritage sites are decreasing.

In the Süleymaniye case, the light grey colour in the analysis indicates positive visibility (the historical buildings can be seen from that site). The dark colour in the first analysis represents that the visibility has decreased due to the small area occupied by the Zeyrek Mosque cultural heritage site and its intensely built environment. The placing of the urban blocks in the vicinity of the Zeyrek Mosque block the visibility of the heritage site.

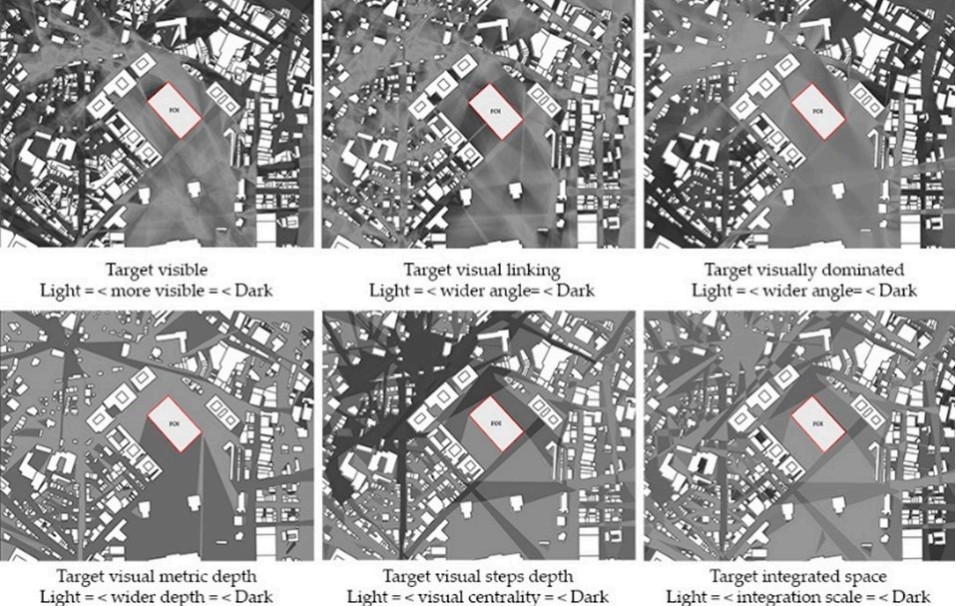

**Figure 9.** Maps of visual metrics of the Süleymaniye Mosque cultural heritage site, generated on the isovist platform [28] by the Author.

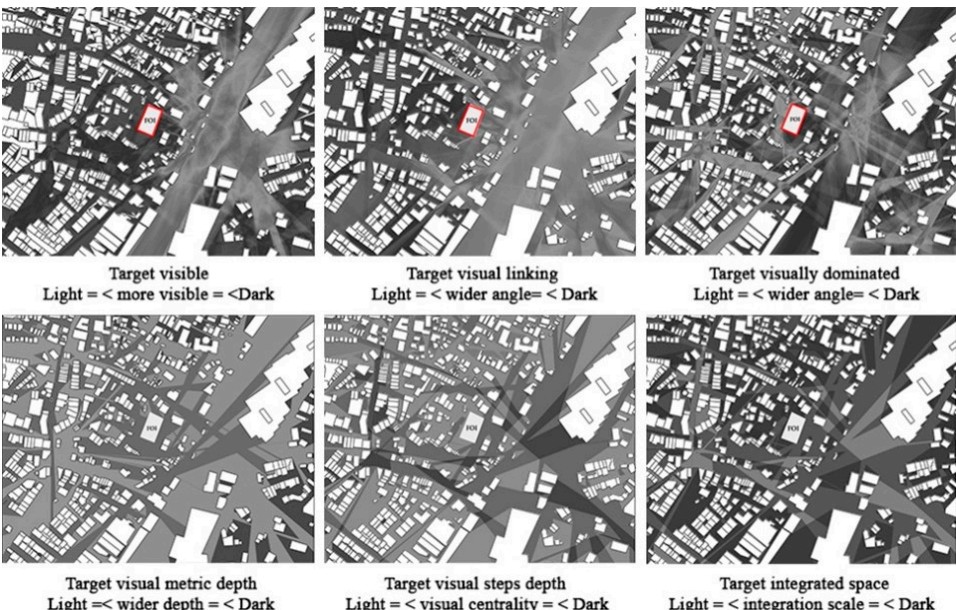

**Figure 10.** Maps of visual metrics of the Zeyrek Mosque cultural heritage site, generated on the isovist platform [28] by the Author.

*6.3. Evaluation of the Visibility Analysis Method—Case Study of the Sulukule Renewal Area*

Located along the Byzantine city walls of the Historical Peninsula, Sulukule is considered the first settlement of the Roma community (Figure 11). According to the information obtained from limited sources, the Roma people arrived from India in 1054 to this region [54].

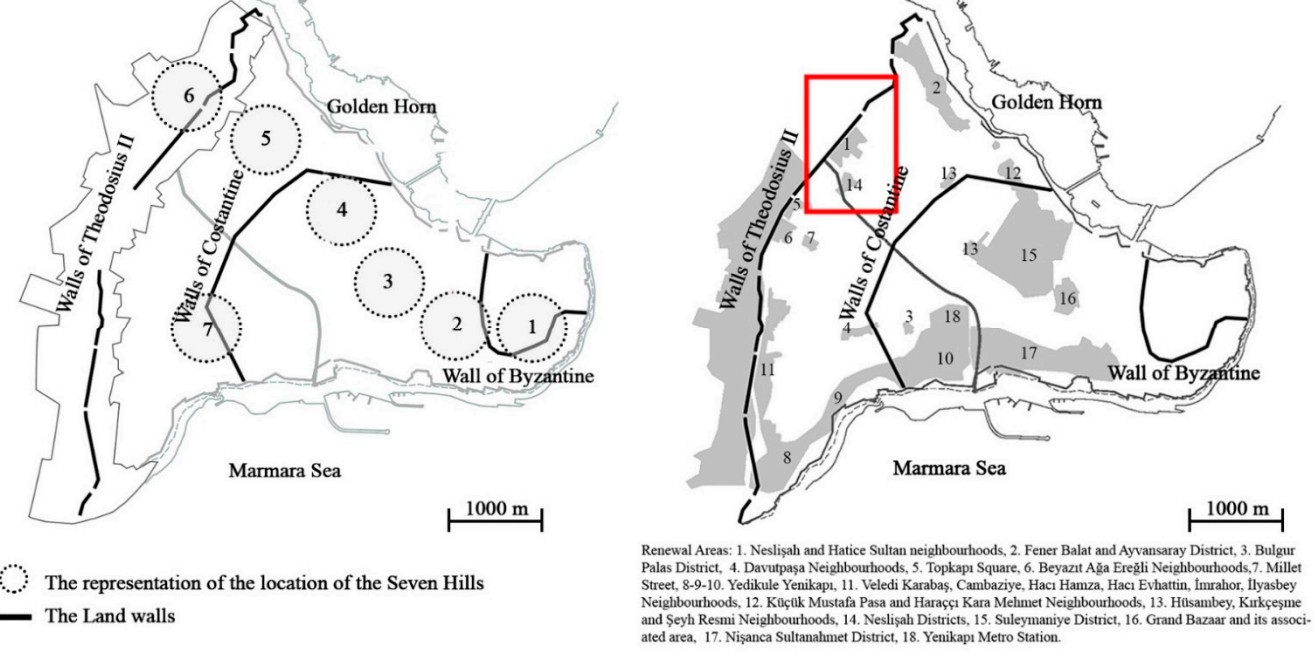

Renewal Areas: 1. Neslişah and Hatice Sultan neighbourhoods, 2. Fener Balat and Ayvansaray District, 3. Bulgur Palas District, 4. Davutpaşa Neighbourhoods, 5. Topkapı Square, 6. Beyazıt Ağa Ereğli Neighbourhoods, 7. Millet Street, 8-9-10. Yedikule Yenikapı, 11. Veledi Karabaş, Cambaziye, Hacı Hamza, Hacı Evhattin, İmrahor, İlyasbey Neighbourhoods, 12. Küçük Mustafa Paşa and Haraççı Kara Mehmet Neighbourhoods, 13. Hüsambey, Kırkçeşme and Şeyh Resmi Neighbourhoods, 14. Neslişah Districts, 15. Suleymaniye District, 16. Grand Bazaar and its associated area, 17. Nişanca Sultanahmet District, 18. Yenikapı Metro Station.

**Figure 11.** Renewal areas in the Historical Peninsula: (1) Sulukule neighbourhood based on [44]; re-illustration by the Author.

People living in Sulukule have undertaken responsibility for the entrance and exit control of the Istanbul land walls and still see these walls as part of their neighbourhood. The urban elements that define the Sulukule district are the narrow streets and the houses with adjoining two-story courtyards shared by the small households surrounding these

streets. Nowadays, the change in the lifestyle of Roma people can be seen [55]. The former residential district, now a ruined and abandoned land, was declared for renovation in 2005. The Sulukule urban renewal project was initiated in 2006 and included 12 blocks, 378 parcels, 10 streets, and 645 architectural structures [56].

The Sulukule case study purposed to determine how the built environment transformed and demonstrates physical and visual integration differences within urban structures.

The changes in the historical landscape (building forms, blocks and patterns) and urban forms of Sulukule were analysed firstly with the urban renewal to identify the visual elements with mapping the area between 2006 and 2020 (Figure 12).

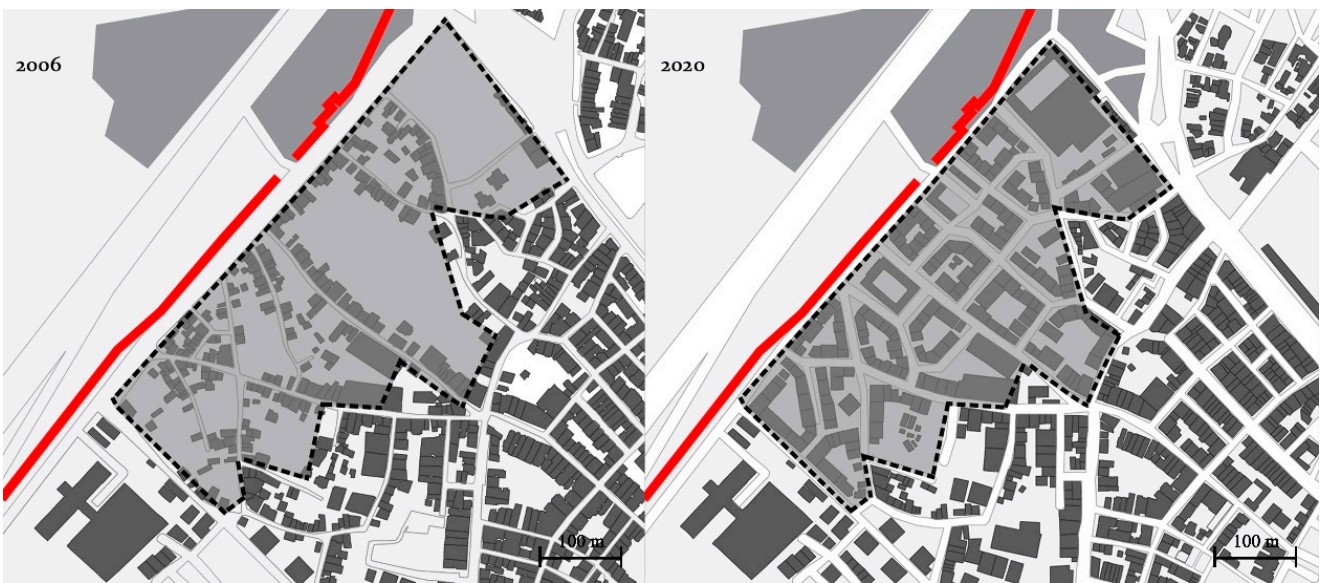

**Figure 12.** Sulukule renewal zone before and after, based on Google Earth; drawing by the Author.

The method applied to the area of 91,000 square meters [57] within the boundaries of the Istanbul Fatih Municipality. UNESCO has chosen this area as it has been evaluated in terms of cultural heritage values, illustrated in the heritage site as "red rectangular".

Located along the Byzantine city walls (marked with red lines), Sulukule is one of the most affected by urban development activities. As the first map shows, the urban form of the Sulukule neighbourhood represented a historical organic structure before the urban renewal process. After the renovation (second map), this structure was completely demolished and transformed into a completely different urban fabric with building blocks. GIS-based mapping analysis enables the characteristics of urban elements to be examined and defined, and how human behaviours in the past and today may have affected the field of visibility and all urban components. In this state, the neighbourhood has lost its identity (tangible and intangible character) and resembles a ghost town built with block buildings. Moreover, the fortification band determined by UNESCO is reduced to half, and the original parcels and street texture have not been retained. The elements belonging to the cultural heritage area have vanished.

One of the important statements of the renovation project is that the Roma Roman settlement that existed for centuries has been demolished, and their social–cultural identity has been separated from the urban context, and the continuity of the community network has been lost. For this purpose, the next stage of the research focuses on the historical values, to compare the visual link that the residents have with historical city walls (Figure 13).

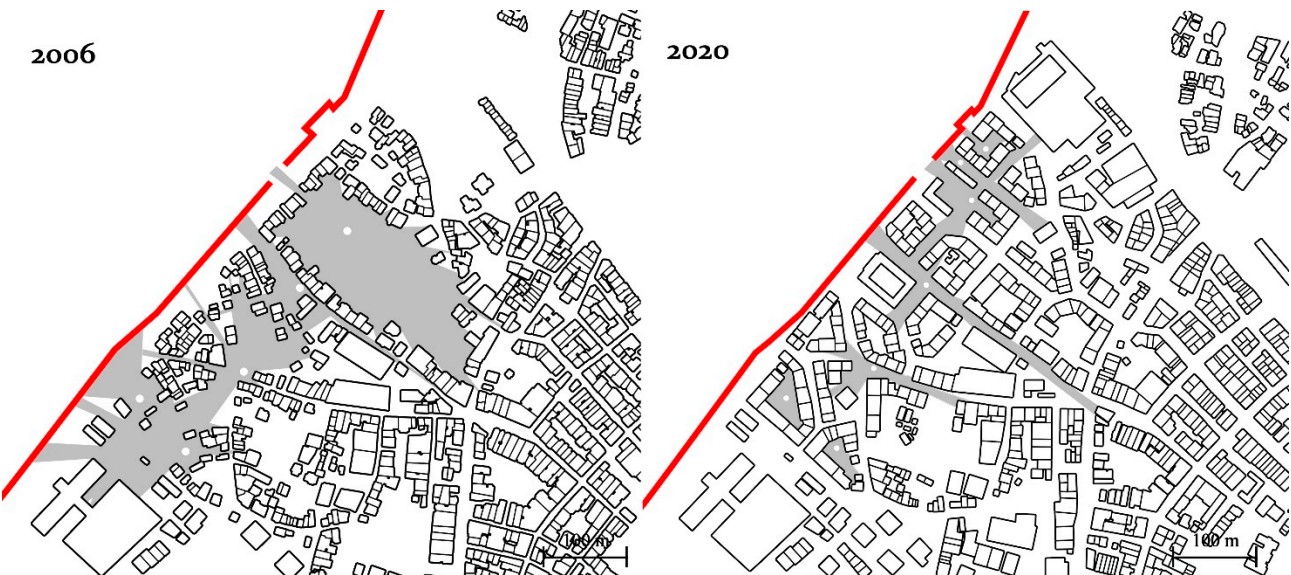

**Figure 13.** Visibility analysis of historical land walls (red) by creating isovist geometries in the Sulukule neighbourhood, generated on the isovist platform [28] by the Author.

The previous analysis (Figure 13) gives information about the importance of the cultural characteristics of the Sulukule neighbourhood to be renewed by preserving the forms of buildings, cultural, and historical structures and landforms in the urban context. Therefore, the visibility analysis focused on the Byzantine walls of the neighbourhood consisting of 12 plots and 378 parcels. The analysis provides an opportunity to understand and discover the changes made in the historic district with its cultural values.

The analysis determines how the visibility of the historic walls has changed from the visual perspective of people who have cultural interactions with the Byzantine heritage in the neighbourhood as the UNESCO's land walls arrangement. Moreover, in most cases, the isovist model [17] is used to define the range of visibility in urban environments, while a viewshed of a topographic dataset in the region is visible from any location around a given observational point [58]. The analyses are applied in heritage regions to determine the visibility of the field of interest (land wall), which can be seen by an observer.

The effects of the isovist visibility modelling were analysed visually. Table 1 shows the number of areas visible before the regeneration process from the same point of view as an observer. The results of the analysis show quantitative evidence of the unpreserved historical identity; of the vanished value of the land walls; and the destroyed cultural heritage visibility. Furthermore, the results of visibility properties indicate spatial and social characteristic destruction of the historic neighbourhood; the identity, quality, and unique character of the heritage site has changed.

**Table 1.** Visible length (meters) of the land walls during the 2007–2020 time period (Source: the Author).

| Period | 2007 | 2020 |
|---|---|---|
| Non-visible | 364.968 | 482.7486 |
| Visible | 187.162 | 69.3814 |

## 7. Discussion

The results of the current study analysis show that the planned/unplanned developments, poor design neighbourhoods, and constantly densely-built environment constitute a distinctive visual/perceptual occupation on the historical, functional, and known values of cultural heritage sites. The urban renewal projects and gentrification efforts increased after 2006 and intensified the buildings in the immediate vicinity of historic icons. These

decisions led to the loss of spatial difference and originality and ignored principles of preserving the integrity of monumental structures. The visibility analysis measurements and isovist models were used to describe visual and configurational properties, visual occupied fields and spaces, and observer behaviour in the heritage space. Visible or occupied visible conditions were validated through visibility graph analysis and comparative visual configurational properties of the spaces by observers' positions. It was observed that there is a high correlation between the visible field and the configurational properties of space. Therefore, to obtain object-level results (FOI)—the visual preservation of an element—it is recommended to take into account the spatial configuration of all elements within the designated area. However, there is a requirement to consider all possible positions of the observer in the given area. The question thus emerges whether the spatial configuration of other elements should be ignored while intending to reveal an element. Therefore, there was a requirement to incorporate the analytical potential of the space syntax and the topology of space into the methodology in order to focus on the geometric and topological properties of the built form in order to find and understand the interrelationships between the differences. On the other hand, it is difficult to track solutions by using GIS-based analysis to identify/predict the constructed and natural features of the environments. The method approach also uses the idea of exploring different possible forms and configurations to understand the interrelationships of factors that lead to visitors' choices. In this context, the research is seeking new contributions to the methodology (visualization/simulation tools/approaches). The idea includes deep learning/understanding the performance of spaces for predicting and improving the interdependence of possible geometric parameters that may arise in the geometry of the space during the renovation and development process. In future research, the simulation of urban space/urban design projects in the early design/planning may remove the limitations of the study in establishing the perceptual behaviour with the information of the built environment.

## 8. Conclusions

The visibility and comprehensibility of heritage sites play a significant and inclusive role in defining the character of the heritage patterns. However, seeing the built environment only in terms of its historical values leads to limited information in terms of seeing different dimensions (changes made in the built environment). Understanding the visible whole (information gathered about all aspects of the urban heritage pattern) and deciding which elements will fit or be included in heritage sites helps us to see the balance between what is planned and what is not. Detailed visibility graph analysis reveals the connection between the urban structure (plan) and the built environment (architecture); it is important to include the configurational analysis and obtain more comprehensive information.

The implemented methodological framework represents a contribution to how both tangible and intangible elements of heritage sites are designed to preserve historic character.

The current study showcases a monitoring/perception tool to evaluate multiple strata of heritage sites to be preserved and developed in urban systems. It is a methodological approach based on the analyses of all elements to reveal the spatial order of heritage sites with visibility analysis.

The main purpose of the study is to evaluate urban elements to be protected by modelling both the heritage environment and the heritage elements according to the visibility criteria. The ultimate purpose of the studies is to evaluate urban elements to protect by modelling both the heritage environment and the heritage elements according to the visibility criteria. In the case of Süleymaniye and Zeyrek, several questionable influences have emerged in the visibility of heritage elements. The concluded heritage elements are in the close vicinity of the listed building. Some of the surrounding buildings are not part of the visibility elements because they are far from or fall outside of the protective zone of the listed buildings. In the case of the renewal site, the built environment and planned urban development trigger the invisibility of heritage elements within the area and visually affect the heritage landscape and harm the protected heritage areas. Urban

development around urban heritage sites may affect the heritage landscape visibility; (existing or future) planning guidelines should suggest the permissible building height of newly constructed buildings around the protection zone. The VGA results formed part of a visibility function that can prioritize information from the current observation position regarding features of interest. The findings present a step towards addressing the issues concerning the importance of protected areas and heritage sites by highlighting critical debates on urban environment evaluation and visibility.

**Funding:** This research received no external funding.

**Institutional Review Board Statement:** Not applicable.

**Informed Consent Statement:** Not applicable.

**Data Availability Statement:** Not applicable.

**Conflicts of Interest:** The authors declare no conflict of interest.

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
