# Peer review of "Visibility Model of Tangible Heritage. Visualization of the Urban Heritage Environment with Spatial Analysis Methods"

_heritage, doi:10.3390/heritage4030122_

Round 1

Reviewer 1 Report

The manuscript proposes “an innovative, yet adaptive way to define and preserve the heritage sites and their elements”.

A visibility analysis should be performed, in order to obtain maps presented in the sections 3 and 4.

Furthermore, the author states: “The purpose of the current study is the evaluation of the identification of the elements to be protected, by modelling both the heritage environment and the heritage elements according to the visibility criteria.”

Later the author states: “The main purpose of the current research is to evaluate the various methods of visibility analyses combining the constituent elements of heritage sites, and possible/already established design and planning indicators in their integration with the built environment”.

The paper sounds scientifically weak.

No definition of the used figures of merit is given. No quantitative examples are shown. Furthermore, only 2D analyses are performed.

Some elaborations are meaningless. For example. in figure 6 the criteria adopted for the choice of the points used for the calculation of the isovist rays are not explained. A slight shift in the position of these points leads to completely different results.

The same remark can be done for figure 8.

With reference to table 1, we must observe that the lenght of visible and not visible Land Walls is approximated to a tenth of millimeter!!

Most important, a reader cannot understand from which point(s) the walls are visible or not.

Author Response

Dear Editor,

Thank you for your comments. I have gone through your comments carefully and tried my best to address them one by one. I hope the manuscript has been improved accordingly.

Below I provide the point-by-point responses.

Kind Regards

Elif SARIHAN

Reviewer 2 Report

The paper is well written and well structured. 

Author Response

Dear Editor,

I appreciate you, reviewing my paper and providing valuable comments. The author welcomes further constructive comments if any.

Kind Regards

Elif SARIHAN

Reviewer 3 Report

please review the punctuation between ; and ,

Figure 1 please change with higher resolution, at least 300 dpi. enlarge the characters.

Is the legend of Figure 1 correct? the two representations of left and should be reported

Figure 2 please change with higher resolution, at least 300 dpi. enlarge the characters.

line 197 the bibliographic reference is missing

line 216 add "is"

Figure 3: please change with higher resolution, at least 300 dpi. how should the sequence be read? is not specified. The meaning of the various colours is not written

Figure 4: it is necessary to introduce a scale bar for the maps. The writings are too small. what are the circles with the numbers? it is not specified. It is not easy to understand. It is necessary to add a legend with the features reported.

Paragraph 3.1 and 3.2. These paragraphs are very important but they are not well explained. According to me, it is necessary 
to integrate the contents of these two paragraphs by making them more descriptive and reduce the different repetitions that are in the introduction

Figure 5: does the content of the figure refer to the previous one? it is not clear

Figure 6: why to compare two different analysis on two different sites without reporting both results? this passage by the author is not clear

Figure 7: please change with higher resolution, at least 300 dpi. What are a) and d)?

line 386: which software? which algorithm? please explain

in figure 8 are introduced the results from isovist software, not well explained before. How can we read these results? please explain. What are the considered areas? are they reported above? Are the images related to the previous? it is not clear

Paragraph 5.1 report the same picture reported before with illegible writings and different colours. The same image is shown over and over again but it is not clear why. If there are more study areas, just put a single image highlighting the different study areas, a single image that is a reference for the reader.

lines 530-541 report information that according to me should be inserted elsewhere. For example, the author should write a paragraph only on the study areas with clear goals.

lines 555-556 where are the changes?

the results are too few, only table 1?

Author Response

Dear Editor,

I appreciate you, reviewing my paper and providing valuable comments. It was your valuable and insightful comments that led to possible improvements in the current version. The author has carefully considered the comments. I hope the manuscript after careful revisions meet your high standards. The author welcomes further constructive comments if any.

Below I provide the point-by-point responses.

Kind Regards

Elif SARIHAN

Reviewer 4 Report

The presentation of methodological and technical aspects is very extensive almost until page 14. Only the last  5 pages are dedicated to the comprehension of the concrete case study and the meaning of the entire methodology. Although authors aim at  clear illustration of methodology, the overall presentation is sometimes redundant and reader has some difficulties in maintaining  the attention.   The conceptual background (integrity, value, territory, landscape, design, ) is extensively  illustrated. The discussion over the fundamental concepts is  complex, but not crystalline in logic, so  it is sometimes difficult to comprehend. Maybe, for the importance of the approach , it could appear less descriptive and  intellectually more challenging. There are  sufficient arguments and information also for other researchers that are willing to reproduce the described methodology as well as reference indications necessary to support the proposed methods.

Author Response

Dear Editor,

I appreciate you, reviewing my paper and providing valuable comments. It was your valuable and insightful comments that led to possible improvements in the current version. The author has carefully considered the comments. I hope the manuscript after careful revisions meet your high standards. The author welcomes further constructive comments if any.

I am grateful for reviewer insightful comments on my paper.

Kind Regards

Elif SARIHAN

Reviewer 5 Report

Dear author,

Please find the following comments and recommendations: 

-In the abstract, the author must justify the importance of the study by referring to the heritage damage registered in the Sulukule neighbourhood as a result of the regeneration process implemented in 2006.

-The sentence "Heritage sites continue to exist in the complexity of contemporary cities as remembrances of the past;" (Row 28) requires the inclusion of at least one citation;

-The sentence: "According to the definition of historic urban landscapes, the protection of cultural  heritage sites plays an essential role in protecting the built environment of these areas [5] " (Rows 36,37) -  there is a repetition, protecting can be replaced by preserving

-The sentence: "In contrast, various obstacles prevent the visible scene of such heritage sites in the historic urban  landscape" (Rows 59,60) requires the inclusion of one or two  citations already used by the author;

-The caption of the Figure 1: it is mentioned name of the author: Philip Thiel which it has to be followed by the corresponding citation number;

-Rows: 197-198 "and the research uses the methods of Bartie and Lopes to determine the object or size that obstructs the visibility conditions of the field of interest (heritage target) " - to include citation number at the end of the sentence or after the names of the authors;

-Row 233: "And thus, it makes it possible to make more detailed inferences to perceptual attributes on the spatial level" -  to avoid repetition, please find a synonym for the second word;

-Rows 355-356: "Zeyrek district known as the fourth hill of Istanbul and recognized as  355 the monastery zone during the early Byzantine period." - requires the inclusion of a citation;

-Results sections: although the results are presented in detail, most of them are included in the Case study section. So, the following sub-sections must be included in the Results section:

 4.1. Visibility graph analysis (VGA) model - The Spatial Connection with Urban Environment

4.2. Mapping the Visual Configurations of Spaces

 including also the Section 5 Implementation of Method of the Renewal Areas

5.1. Evaluation of the visibility analysis method - case study Sulukule renewal area.

The case study section must refer to the description of the importance of the two mosques analysed and the areas adjacent them which still preserve historical and architectural values. Otherwise, the results section seems fragmented.

-The discussion section is missing. In the discussion section, several aspects must be included : the significance of the results, the author can highlight similar results identified in previous studies; the limitations of the study and future research directions that could eliminate these limitations.

-In the conclusions section, the author uses citations which should be avoided.

Author Response

(The authors gave the same response as above.)

Round 2

Reviewer 1 Report

Author didn't address adequately the remarks of the reviewer

Author Response

Dear Editor,

I appreciate you, reviewing my paper and providing valuable comments. The author welcomes further constructive comments if any.

Kind Regards

Elif SARIHAN

This manuscript is a resubmission of an earlier submission. The following is a list of the peer review reports and author responses from that submission.